# Tea Quality of the Mysterious “Dahongpao Mother Tree” (*Camellia sinensis*)

**DOI:** 10.3390/foods13101548

**Published:** 2024-05-16

**Authors:** Jianghua Ye, Qi Zhang, Mingzhe Li, Yuhua Wang, Miao Jia, Lei Hong, Yiling Chen, Xiaomin Pang, Xiaoli Jia, Haibin Wang

**Affiliations:** 1College of Tea and Food, Wuyi University, Wuyishan 354300, China; jhye1998@126.com (J.Y.); zhangqi1113@126.com (Q.Z.);; 2College of Life Science, Longyan University, Longyan 364012, China; 3College of JunCao Science and Ecology, Fujian Agriculture and Forestry University, Fuzhou 350002, China

**Keywords:** *Camellia sinensis*, Dahongpao mother tree, metabolomics, odor characteristics, taste characteristics

## Abstract

The quality of the Dahongpao mother tree (*Camellia sinensis*) remains a mystery to this day. In this study, for the first time, the differences between the Dahongpao mother tree (MD) and Dahongpao cuttings (PD), in terms of odor characteristics and taste characteristics were analyzed by metabomics. The results showed that MD had stronger floral, fruity, green, and woody odor characteristics than PD, and that the contributions were mainly from dihydromyrcenol, methyl salicylate, 2-isobutylpyrazine, 1,6-dihydrocarveol, gamma-terpineol, and linalyl acetate. Further, fresh and brisk taste and mellowness taste characteristics of MD were significantly higher than PD, with contributions mainly from amino acids and derivatives and organic acids. Secondly, bitterness taste characteristics of PD were significantly higher than MD, with contributions from phenolic acids, flavones, and flavonols. This study preliminarily unraveled the legend of the superior quality of the Dahongpao mother tree, and also provided an important reference for the breeding of tea-tree cuttings.

## 1. Introduction

Wuyi Mountain is a World Cultural and Natural Heritage and a World Biosphere Reserve. At the same time, China’s Wuyi Mountain is an essential tea-producing area, as it belongs to a typical Danxia landform, thus forming tea with a special flavor—Wuyi Rock Tea. The Dahongpao mother tree is the tea goddess of Wuyi Mountain and, with a history of nearly 400 years, it reflects the culture and history of Wuyi-rock-tea cultivation. Dahongpao mother tree refers to the Dahongpao mother tree that grows in the Jiulongke Scenic Area of Wuyi Mountain, with three trees and six plants [1]. In the 1930s, the local government sent soldiers to guard the Dahongpao mother tree, and, since the founding of the People’s Republic of China, relevant departments still employ farmers to look after the tree year after year. In 2000, Wuyi Mountain was successfully declared a World Natural and Cultural Heritage site, and the Dahongpao mother tree was listed as a key conservation target in the Regulations for the Protection of World Cultural and Natural Heritage in Wuyi Mountain, Fujian Province [2]. In 2003, Wuyishan Municipal Government took out a 100 million RMB product liability insurance policy with the People’s Insurance Company of China for the six existing Dahongpao mother trees. On 3 May 2005, 20 g of the tea from the most recent harvest from the Dahongpao mother tree of Wuyi Mountain were officially presented to the National Museum of China by the Wuyi Municipal People’s Government [3]. The Wuyishan Municipal Government decided, from 2006, that Dahongpao mother tree stopped from being harvested, only by the tea professional and technical personnel of Dahongpao mother tree to implement scientific management. Since then, the tea of the Dahongpao mother tree has not been on the market.

Prior to the cessation of harvesting, the tea of the Dahongpao mother tree was sought by tea consumers everywhere due to its scarcity and unique quality [4]. In 1998, at the first China Wuyishan Dahongpao Tea Culture Festival, 20 g of the tea from the Dahongpao mother tree were auctioned for as much as 156,800 RMB. In 2005, at the seventh Wuyi Mountain Hongpao Festival, 20 g of Dahongpao mother tree tea was auctioned for as much as 208,000 RMB. At this point, Dahongpao Mother Tree tea became the most expensive tea in the world. Dahongpao mother tree tea is a rare and uncommon tea. In the 1980s, tea scientists used asexual reproduction cuttings of the “Dahongpao mother tree” and developed Dahongpao into a famous tea brand for more than 40 years. However, on the Dahongpao mother tree and Dahongpao cuttings in the quality of whether there is a difference, has never seen any scientific research reports, but only in the folklore there are some legends. For example, Xiang [5] argued that only the Dahongpao mother tree is pure, but it is hardly circulated in the market. Ng et al. [6] concluded that cuttings are asexually propagated and the quality of Dahongpao cuttings would not change much. DeBernardi [7] argued, from a cultural point of view, that the Dahongpao mother tree creates a link between the present and the past, and that the modern Dahongpao cuttings are something new, a product of scientific innovation, an icon of creativity, and a model of consumption for the new elite. It can be seen that the scarcity of the Dahongpao mother tree has left a large number of legends in folklore. Is there a difference between the quality of the Dahongpao mother tree and Dahongpao cuttings? This has become an unsolved mystery in the tea world. However, the meticulous analysis and comprehensive understanding of the Dahongpao mother tree’s exceptional quality characteristics hold considerable historical significance for the future progressive expansion and diversification of the Dahongpao industry, and even for the overall development of the Wuyi-rock-tea industry.

Volatile compounds and nonvolatile metabolites present in tea leaves have a significant impact on the formation of tea qualities [8,9]. Among them, volatile compounds are mainly used to evaluate the odor characteristics of tea [10], while nonvolatile metabolites are mainly used to evaluate the taste characteristics of tea [11]. Accordingly, this study, for the first time, took the Dahongpao mother tree (MD) and the first batch of asexually propagated Dahongpao cuttings (PD) in the same area in 1980s as materials, and used the joint analysis of broad-targeted metabolomics and volatile metabolomics to explore the differences in volatile compounds and nonvolatile metabolites between MD and PD, and further screened for characteristic compounds and their contributions to the formation of the odor and taste characteristics, so as to make the difference in the qualities of MD and PD clear, and to preliminarily reveal the legends of the excellent qualities of the Dahongpao mother tree, with the aim of providing references for the sustainable development of the Wuyi-rock-tea industry.

## 2. Materials and Methods

### 2.1. Materials

The main reagents utilized in the experiments, acetonitrile, formic acid, n-hexane, and methanol, were chromatographic grade and sourced from Sigma (St. Louis, MO, USA). All conventional reagents, except for the few mentioned specifically, were of analytical grade and were procured from Sinopharm (Beijing, China).

The objects of this study were the Dahongpao mother tree (MD) and the first asexually propagated Dahongpao cuttings (PD) in the same area in the 1980s. The experimental sampling site was located in Jiulongke Scenic Area, Wuyishan City, Fujian Province, China (117°57′19.098″ E, 27°40′17.8212″ N). Tea-tree varieties were all Dahongpao tea trees, of which MD was about 390 years old and PD was about 40 years old. In May 2023, three leaves and one core of MD and PD tea trees were collected and promptly preserved in liquid nitrogen for determination of volatile compounds and nonvolatile metabolites, with three independent replicates of each sample.

### 2.2. Determination of Volatile Compounds in Tea-Tree Leaves

The collected fresh tea leaves were ground into powder using liquid nitrogen, and vortexed and mixed separately. Then, 0.5 g of the ground sample was placed in a headspace vial (Agilent, Palo Alto, CA, USA), 10 mL of saturated NaCl solution was added, and volatile compounds were collected using a fully automated headspace solid-phase microextraction (HS-SPME) with three independent replicates per sample. HS-SPME extraction was performed by shaking the headspace vial containing the sample at 60 °C for 5 min, then immediately inserting a 120 µm extraction head (DVB/CWR/PDMS, Agilent, Palo Alto, CA, USA) into a headspace vial and extracting for 15 min. The extraction head was pre-treated in a Fiber Conditioning Station at a temperature of 250 °C for a duration of 2 h before it was utilized.

Upon completion of volatile compound collection, volatile compounds were detected and quantified using Agilent gas chromatography (8890B, Agilent, Santa Clara, CA, USA) coupled with mass spectrometry (7000D, Agilent, Santa Clara, CA, USA). Before analysis was initiated, the extraction head of the collected sample was desorbed at a temperature of 250 °C for 5 min. This was done at the inlet of the gas chromatograph. The capillary column used in the gas chromatograph was DB-5MS (30 m × 0.25 mm × 0.25 μm, Agilent, Santa Clara, CA, USA), and the experiment used a high-purity helium carrier gas (purity > 99.999%) at a constant flow rate of 1.2 mL/min. The gas was introduced into the GC at a temperature of 250 °C and a non-split injection was used, with a solvent delay of 3.5 min. The heating sequence was as follows: (I) it remained at 40 °C for 3.5 min; then, (II) the temperature was raised by 10 °C/min until it reached 100 °C; (III) it was raised by 7 °C/min until it reached 180 °C; and finally, (IV) it was raised by 25 °C/min until it reached 280 °C, where it remained for 5 min. The conditions for the mass spectrometry analysis included the use of electron bombardment as the ionization technique, a source temperature of 230 °C, a quadrupole temperature of 150 °C, a temperature of 280 °C for the mass spectrometry interface, an electron energy of 70 eV, and a scanning mode in the selected ion detection mode.

For each compound, the qualitative and quantified analytical methods involved selecting a specific quantitative ion, and an additional two to three qualitative ions. During the comparison process using the NIST20 mass spectrometry database, the ions under test were detected in time intervals according to the order of peaks, and the substance was qualitatively determined to be the substance of interest, provided that the retention time of the detected ions was in agreement with the standard reference value, and that the selected ions also appeared in the mass spectra of the samples after subtraction of the background interference [12]. After qualitative results were determined, the quantitative ions were carefully selected for integration and correction to ensure the accuracy and precision of the quantification process [13].

### 2.3. Determination of Nonvolatile Metabolites in Tea-Tree Leaves

The collected tea fresh leaves were ground into powder using liquid nitrogen, and vortexed and mixed separately. Firstly, 50 mg of ground tea powder was taken and then 1200 μL of pre-cooled methanol solution (70%) at −20 °C was added. The solution was then vortexed and shaken for 30 s at 30 min intervals for a total of 6 times. Next, it was put in 12,000 rpm/min centrifugation, and the supernatant was transported via a microporous filter membrane (0.22 μm) for subsequent analysis of metabolite content using ultra performance liquid chromatography mass spectrometry (UPLC, ExionLC™AD, AB SCIEX, BSN, Framingham, MA, USA; MS, Applied Biosystems 6500 Q TRAP, Foster City, CA, USA) with three replicates per sample [14].

The UPLC column was SB-C18 (1.8 µm, 2.1 mm × 100 mm). The mobile phases were composed of phase A (ultrapure water containing 0.1% formic acid) and phase B (acetonitrile containing 0.1% formic acid). The gradient elution mode was set such that the phase A/B ratio was 95/5 at 0 min, the phase B ratio linearly increased to 95% within 9.00 min then remained constant for 1 min. At 10~11 min, the phase B ratio decreased to 5% and remained constant for 3 min. A temperature of 40 °C was selected for the UPLC column. The optimal injection volume of 2 μL was also established, with a flow rate of 0.35 mL/min ensuring an efficient analysis. The experimental procedure for the MS analysis included the following steps: (I) Using an electrospray ionization source, ions were generated at a temperature of at 500 °C. (II) The MS detection was conducted in both positive and negative ion modes. The specific voltage for positive ion mode was 5500 V while it was −4500 V for negative ion mode. (III) For ion sources I and II, the operating pressures were set to 50 and 60 psi, respectively. The air curtain gas pressure was set to 25 psi. (IV) Tandem quadrupole mass spectrometry was used in multiple reaction monitoring mode. (V) The declustering potential and collision energy of each multiple reaction monitoring ion pair were optimized. The specific multiple reaction monitoring ion pairs were then monitored based on the eluted metabolites in each period.

According to the secondary mass spectrometry information, the NIST20 mass spectrometry database was provided for comparison and qualitative analysis of nonvolatile metabolites. The comparison process systematically eliminated isotopic signals, repetitive signals for K^+^, Na^+^, and NH_4_^+^, and repetitive signals from fragment ions that themselves represent larger molecular weight substances [15]. After qualitative analysis was completed, characteristic ions for each metabolite were screened by tandem quadrupole mass spectrometry. In the detector, the signal intensities of the characteristic ions were obtained. The downstream mass spectrometry files of the samples were opened with MultiQuant software (Version 3.0.3). After performing integration and correction work on the chromatographic peaks, each chromatographic peak area represented the relative content of the corresponding metabolite. Finally, all the chromatographic peak area integration data were exported and saved [16]. The key step in the analysis was comparison of the content of all metabolites detected in different samples. For this purpose, the chromatographic peaks of each metabolite, detected in various samples, were corrected based on the retention time versus peak shape information. This laborious process ensured that the obtained results were both precise and accurate, thus enabling the accurate identification and quantification of the different metabolites present in each sample.

### 2.4. Statistical Analysis

Microsoft Excel 2020 was used to perform preliminary statistical analysis of raw data. Rstudio software (version 4.2.3) was utilized for graphic production of the post-statistical data. The R package used for the box plot was gghalves 0.1.4. The R package used for principal component analysis (PCA) was ggbiplot 0.55. The R packages used for the scatter radar map were ggradar 0.2, readxl 1.4.3, ggplot2 3.5.0, tidyverse 2.0.0, scales 1.3.0, and cowplot 1.1.3. The R package used for the volcano map was ggplot2 3.5.0. The R package utilized for technique for order of preference by similarity to ideal solution (TOPSIS) analysis was dplyr 1.1.4. The R packages used to construct the orthogonal partial least squares discrimination analysis (OPLS-DA) model were ropls and mixOmics. The R package used for the bubble feature map was ggplot2 3.4.4. The R packages used for the odor and taste wheels were complexHeatmap version 2.16.0, vegan version 2.6.4, circlize version 0.4.15, and RColorBrewer version 1.1.3. The R packages used for the petal plot were ggplot2 3.5.0, ggprism 1.0.4, and ggthemes 5.1.0.

## 3. Results and Discussion

### 3.1. Volatile Metabolomics Analysis of Tea-Tree Leaves

The content and composition of volatile compounds play a significant role in the development of tea’s aroma characteristics. By accurately analyzing these compounds, the strength of the odor characteristics of tea can be accurately determined, subsequently enabling deeper analysis and understanding of tea’s aroma characteristics [17,18]. This study used volatile metabolomics techniques to determine volatile compounds in leaves of the Dahongpao mother tree (MD) and Dahongpao cuttings (PD), and showed (Figure 1A, Appendix A) that a total of 602 volatile compounds were identified in MD and PD, and there was no significant difference in the total number of volatile compounds between MD and PD (*p* > 0.05). Principal component analysis of the contents of 602 volatile compounds revealed (Figure 1B) that there was a clear distinction between MD and PD, and the two principal components effectively distinguished MD from PD, with a total contribution of 90.90%. It can be seen that although the difference between MD and PD was not significant in the total amount of volatile compounds, they were significantly different in the content of different volatile compounds. Therefore, the detected volatile compounds were further classified into 16 categories. Of these, six categories were more relevant to MD, namely aldehyde, nitrogen compounds, heterocyclic compounds, terpenoids, aromatics, and others, while three categories were more relevant to PD, namely sulfur compounds, hydrocarbons, and ester (Figure 1C). Analysis on the content of 16 categories of volatile compounds found (Figure 1D) that sulfur compounds, aromatics, terpenoids, heterocyclic compounds, nitrogen compounds, and others had significant differences between MD and PD. It was evident that a significant difference existed between MD and PD in the content of different categories of volatile compounds, and these differences may lead to a difference in aroma intensity and odor characteristics of the tea.

### 3.2. Screening for Characteristic Volatile Compounds and Their Odor Characteristics

Odor characteristics of different volatile compounds are different; therefore, screening to obtain characteristic volatile compounds with significant differences among different samples is important for further analyzing the differences in odor characteristics and their aroma intensities among different samples [13,19]. On the basis of the above analysis, a volcano diagram was used in this study to further screen volatile compounds that had significant differences in MD and PD, and the results showed (Figure 2A) that a total of 192 volatile compounds that had differences were screened, of which 136 were significantly increased and 56 significantly decreased in MD compared with PD. The 192 volatile compounds were further classified into 14 categories. TOPSIS analysis showed (Figure 2B) that only 7 of the 14 categories had a weighting of 10% or more in distinguishing MD and PD, namely ester (77.95%), terpenoids (62.58%), heterocyclic compound (18.57%), ketone (14.65%), alcohol (14.40%), hydrocarbons (12.16%), and aldehyde (11.75%). The OPLS-DA model of MD and PD was further constructed to screen key differential volatile compounds, and the results showed (Figure 2C) that the fit (R^2^Y = 0.999, *p* < 0.005) and predictability (Q^2^ = 0.999, *p* < 0.005) of the OPLS-DA model reached a significant level, and the model was able to effectively differentiate MD from PD, with the total score of 97.96%. A total of 116 key differential volatile compounds were screened by the OPLS-DA model. Then, a bubble feature map further screened characteristic volatile compounds with significant differences, and a total of 57 characteristic volatile compounds were obtained (Figure 2D).

On the above analysis, this study used TOPSIS to analyze the importance of 57 characteristic volatile compounds in the process of distinguishing MD from PD and found that (Figure 2E), only 8 characteristic volatile compounds had a weight of 10% or more, namely methyl salicylate (61.32%), (*E*)-3-hexen-1-ol acetate (49.79%), (*Z*)-3-hexen-1-ol acetate (49.79%), 1,6-dihydrocarveol (17.27%), gamma-terpineol (17.27%), dihydromyrcenol (16.16%), linalyl acetate (13.80%), and 2-isobutylpyrazine (10.93%). Of eight characteristic volatile compounds that are considered essential to odor characteristics of tea, methyl salicylate, (*E*)-3-hexen-1-ol acetate, (*Z*)-3-hexen-1-ol acetate, and 2-isobutylpyrazine mainly have a fruity odor characteristic [20,21,22,23], dihydromyrcenol has a floral odor characteristic [24], 1,6-dihydrocarveol has a green odor characteristic [25], and gamma-terpineol and linalyl acetate have a woody odor characteristic [26,27]. It can be seen that MD and PD differed mainly in fruity, floral, green, and woody odor characteristics, while the difference in the content of volatile compounds of different odor characteristics may lead to the difference in the intensity of different odor characteristics between MD and PD. Accordingly, in-depth analysis of the intensity of odor characteristics of the eight characteristic volatile compounds found (Figure 3A) that the content of dihydromyrcenol, methyl salicylate, 2-isobutylpyrazine, 1,6-dihydrocarveol, gamma-terpineol, and linalyl acetate was significantly greater in MD than in PD, while the content of (*E*)-3-hexen-1-ol acetate and (*Z*)-3-hexen-1-ol acetate was significantly lower in MD than in PD. The combined analysis of odor characteristics showed (Figure 3B) that floral, fruity, green, and woody were significantly greater in MD than in PD. It was evident that although the difference between MD and PD was not significant in the total amount of volatile compounds, different volatile compounds had significant differences in the content, especially the eight characteristic volatile compounds, which contributed to the stronger odor characteristics of floral, fruity, green, and woody for MD than for PD.

### 3.3. Broad-Target Metabolomics Analysis of Tea-Tree Leaves

Nonvolatile metabolites in tea are the main compounds that make up the special taste of tea, and the content of different metabolites directly affects taste characteristics of tea [28,29]. Therefore, analyzing the content of nonvolatile metabolites in tea is important for determining the taste characteristics of tea. In this study, we determined the nonvolatile metabolites in the leaves of the Dahongpao mother tree (MD) and Dahongpao cuttings (PD) using broad-targeted metabolomics, and the results showed (Figure 4A, Appendix A) that a total of 1971 metabolites were identified in MD and PD, while there was no significant difference in the total amount of each between MD and PD (*p* > 0.05). Principal component analysis with the contents of 1971 metabolites revealed (Figure 4B) that there was a clear distinction between MD and PD, and that two principal components effectively distinguished MD from PD with a total contribution of 71.20%. The detected metabolites were further classified into 13 categories. Of these, six categories were more relevant to MD, namely tannins, nucleotides and derivatives, lipids, amino acids and derivatives, alkaloids, and others, while seven categories were more relevant to PD, namely phenolic acids, quinones, organic acids, flavonoids, lignans and coumarins, steroids, and terpenoids (Figure 4C). The content analysis of the 13 categories of metabolites showed (Figure 4D) that flavonoids, lipids, lignans and coumarins, steroids, organic acids, and others were significantly different between MD and PD. It can be seen that although the difference between MD and PD was not significant in terms of the total amount of metabolites, a significant difference existed in the composition and content of different metabolites. This phenomenon may also cause MD and PD to show some differences in taste characteristics.

### 3.4. Screening for Characteristic Metabolites and Their Taste Characteristics

Different nonvolatile metabolites in tea have different taste characteristics, and screening for characteristic metabolites with significant differences in different samples is helpful for further determining the differences in taste characteristics of the samples [30,31]. In the above analysis, a volcano diagram was used to further screen nonvolatile metabolites with significant differences between MD and PD, and the results showed (Figure 5A) that a total of 420 differential metabolites were identified, of which 178 were significantly increased and 242 significantly decreased in MD compared with PD. The 420 metabolites were further classified into 12 categories. TOPSIS analysis showed (Figure 5B) that only 6 of the 12 categories had a weight of 10% or more in distinguishing between MD and PD, namely flavonoids (93.23%), phenolic acids (82.03%), others (18.51%), amino acids and derivatives (17.08%), tannins (16.08%) and organic acids (14.81%).

The OPLS-DA model of MD and PD was further screened for key differential metabolites, and the results showed (Figure 5C) that the fit (R^2^Y = 1, *p* < 0.005) and predictability (Q^2^ = 1, *p* < 0.005) of the model reached the significance level, and the model was able to effectively discriminate between MD and PD with an overall score of 95.77%. A total of 233 key differential metabolites were obtained by the constructed OPLS-DA model. Then, a bubble feature map was further screened for characteristic metabolites with significant differences, and a total of 112 characteristic metabolites were obtained, which could be subdivided into 16 categories (Figure 5D). Based on the above analysis, further analysis of the importance of 16 categories of characteristic metabolites in the process of distinguishing MD from PD using TOPSIS revealed (Figure 5E) that 8 categories had a weight of 10% or more in the process of distinguishing MD from PD, namely phenolic acids (100%), amino acids and derivatives (35.31%), flavones (33.98%), tannin (31.08%), flavonols (31.06%), flavanols (31.02%), organic acids (30.88%), and others (28.06%). The main taste characteristic reported for phenolic acids and tannin is bitterness [32,33]. Flavones, flavonols, and flavanols are also characterized by bitterness [34,35]. Amino acids and derivatives are characterized by fresh and brisk taste [36]. And organic acids are characterized by mellowness [37]. It can be seen that the differences in characteristic metabolites between MD and PD may lead to some differences in taste characteristics, especially in taste characteristics such as bitterness, fresh and brisk taste, and mellowness. This study analyzed characteristic metabolite contents of MD and PD as well as their taste characteristic intensities, and the results showed (Figure 6A) that the content of tannin, flavanols, amino acids and derivatives, and organic acids in MD was significantly greater than that of PD, while the content of phenolic acids, flavones, and flavonols was significantly lower than that of PD. The combined analysis of taste characteristics showed (Figure 6B) that the intensity of fresh and brisk taste and mellowness was significantly greater in MD than in PD, while the intensity of bitterness was significantly lower than in PD. It was evident that although the difference between MD and PD was not significant in the total amount of nonvolatile metabolites, the content of different categories of metabolites was significantly different, especially amino acids and derivatives, phenolic acids, flavones, tannin, flavonols, flavanols, and organic acids, which, in turn, enhanced the fresh and brisk taste and mellowness and reduced the bitterness of MD.

## 4. Conclusions

The quality characteristics of the Dahongpao mother tree have long been a legend of Wuyi rock tea, and it remains a mystery as to what exactly the difference in quality is between it and Dahongpao cuttings. In this study, metabolomics techniques were used to analyze the differences in odor characteristics and taste characteristics between the Dahongpao mother tree (MD) and Dahongpao cuttings (PD) in terms of volatile compounds and nonvolatile metabolites. This study found (Figure 7) that MD had stronger odor characteristics, such as floral, fruity, green, and woody, and stronger taste characteristics, such as fresh and brisk taste and mellowness, than PD. Secondly, the bitterness of MD was significantly lower than that of PD. It can be seen that the composition of volatile compounds and non-volatile metabolites of asexual PD was similar to that of MD, but there was a significant difference in their content, so that MD was significantly superior to PD in both odor characteristics and taste characteristics. This study analyzed, for the first time, the difference in quality between the Dahongpao mother tree and Dahongpao cuttings, and unveiled the legend of the superior quality of the Dahongpao mother tree. Meanwhile, the study also illustrated that although tea-tree cuttings are asexually propagated, they may have some differences in the leaf quality. The results of this study also provide an important reference for tea-tree cultivation and breeding.

## Figures and Tables

**Figure 1 foods-13-01548-f001:**
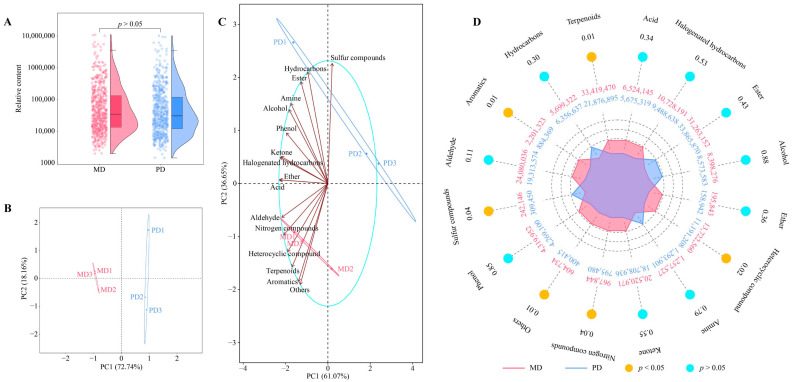
Volatile metabolome analysis of tea-tree leaves. Note: MD: Dahongpao mother tree and PD: Dahongpao cuttings. (**A**) Analysis of the total amount of volatile compounds in tea-tree leaves, (**B**) PCA analysis of volatile compounds in tea-tree leaves, (**C**) PCA analysis of volatile compounds of tea-tree leaves after classification, and (**D**) content analysis of volatile compounds of tea-tree leaves of different categories.

**Figure 2 foods-13-01548-f002:**
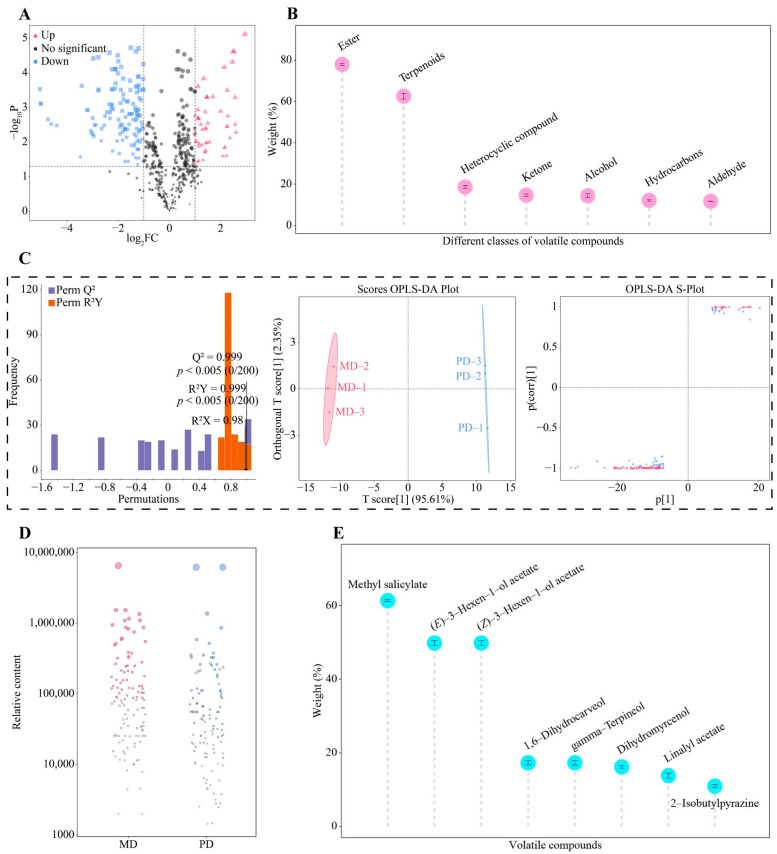
Screening for characteristic volatile compounds. Note: MD: Dahongpao mother tree and PD: Dahongpao cuttings. (**A**) Screening for volatile compounds that were significantly different between MD and PD by volcano diagram, (**B**) weighting analysis of different categories of volatile compounds in the process of distinguishing MD and PD, (**C**) screening for volatile compounds with key differences by OPLS-DA model, (**D**) screening for characteristic volatile compounds by bubble feature map, and (**E**) weight analysis of characteristic volatile compounds in the process of distinguishing MD and PD.

**Figure 3 foods-13-01548-f003:**
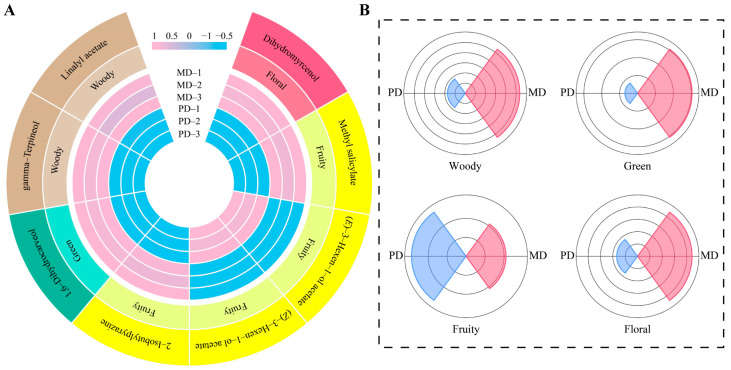
Odor characteristics analysis of characteristic volatile compounds. Note: MD: Dahongpao mother tree and PD: Dahongpao cuttings. (**A**) Content of characteristic volatile compounds and their odor characteristics and (**B**) odor characteristic intensity analysis of characteristic volatile compounds in MD and PD.

**Figure 4 foods-13-01548-f004:**
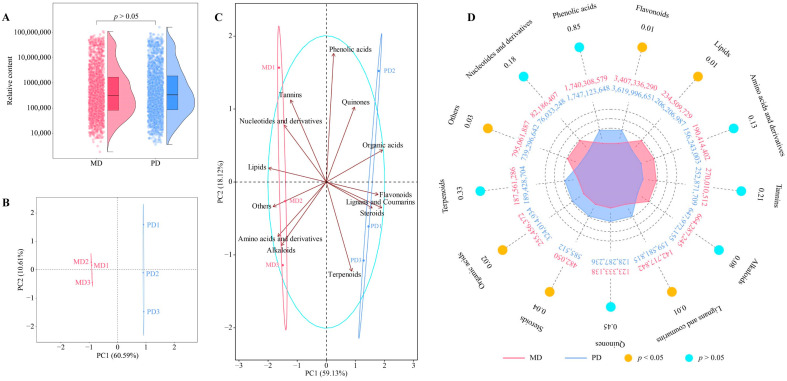
Broad-target metabolomics analysis of tea-tree leaves. Note: MD: Dahongpao mother tree and PD: Dahongpao cuttings. (**A**) Analysis of the total amount of metabolites in tea-tree leaves, (**B**) PCA analysis of tea-tree-leaf metabolites, (**C**) PCA analysis of metabolites in tea-tree leaves after classification, and (**D**) content analysis of metabolites of different categories of tea-tree leaves.

**Figure 5 foods-13-01548-f005:**
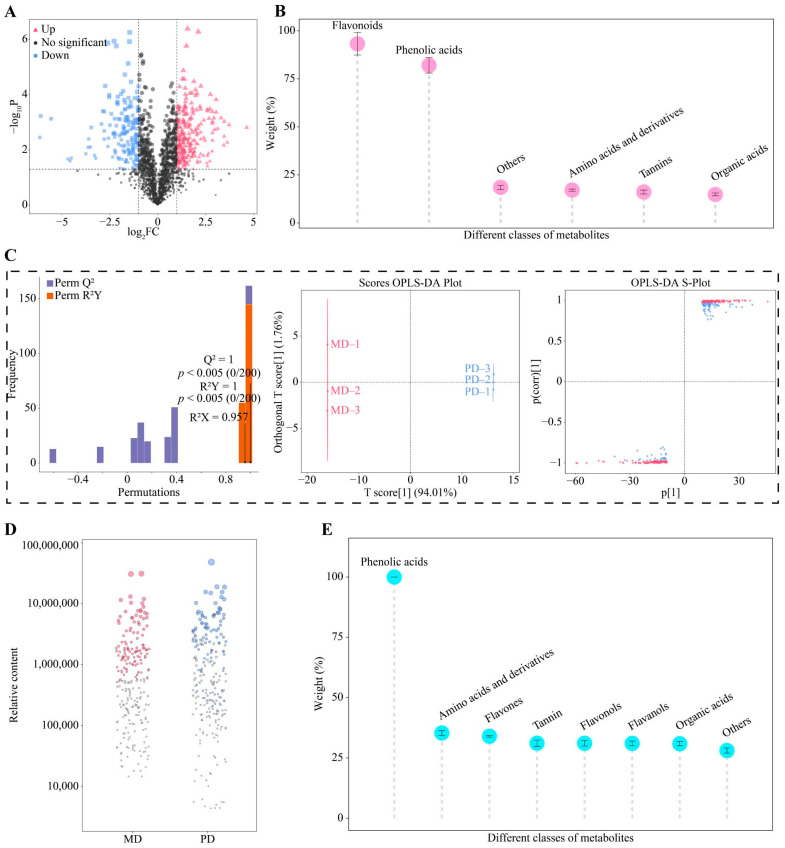
Screening for characteristic metabolites. Note: MD: Dahongpao mother tree and PD: Dahongpao cuttings. (**A**) Screening for metabolites was significantly different between MD and PD by volcano diagram, (**B**) weighting analysis of different categories of metabolites in the process of distinguishing MD and PD, (**C**) screening for metabolites with key differences by OPLS-DA model, (**D**) screening for characteristic metabolites by bubble feature map, and (**E**) weight analysis of characteristic metabolites in the process of distinguishing MD and PD.

**Figure 6 foods-13-01548-f006:**
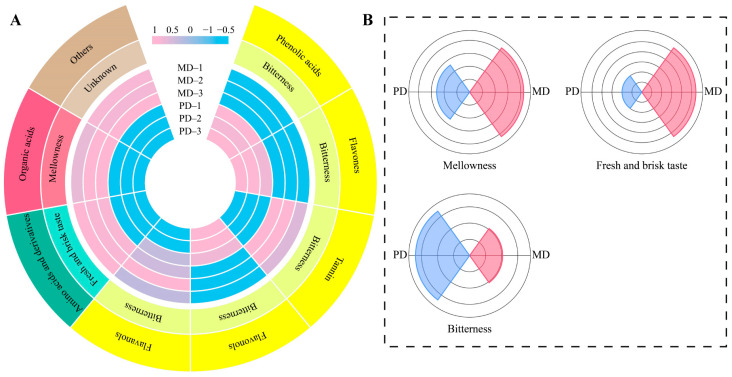
Taste characteristics of characteristic metabolites. Note: MD: Dahongpao mother tree and PD: Dahongpao cuttings. (**A**) Content of characteristic metabolites and analysis of their taste characteristics and (**B**) analysis of taste characteristics intensity of characteristic metabolites in MD and PD.

**Figure 7 foods-13-01548-f007:**
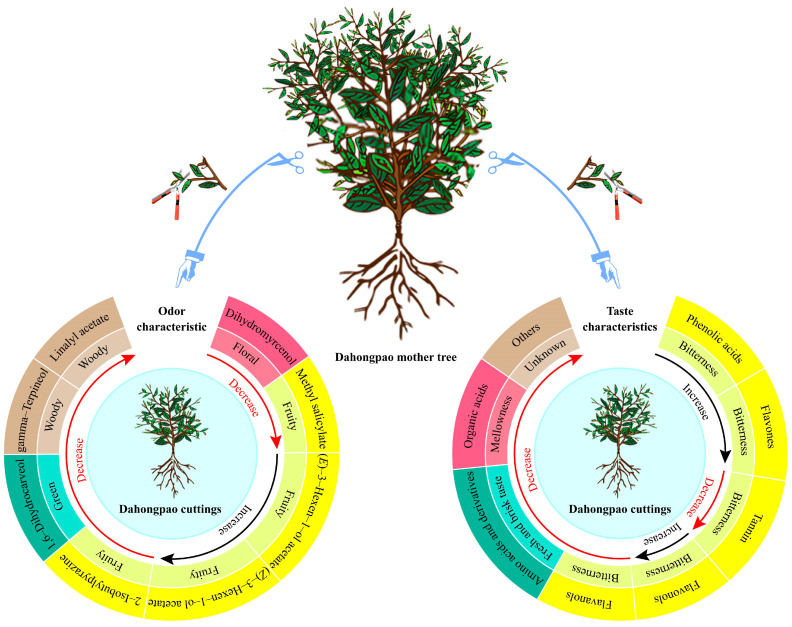
Mechanism of quality change of Dahongpao mother tree and Dahongpao cuttings.

## Data Availability

The original contributions presented in the study are included in the article/Appendix A, further inquiries can be directed to the corresponding author.

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
