# Peer review of "Tea Quality of the Mysterious “Dahongpao Mother Tree” (Camellia sinensis)"

_foods, 2024, doi:10.3390/foods13101548_

Round 1

Reviewer 1 Report

Comments and Suggestions for Authors

The study is interesting, reporting the metabomics differences between Dahongpao mother three and Cutting Dahongpao, in terms of odor and taste characteristics. The study contributes to the development of knowledge regarding this plant species that is so important in Chinese culture.

The research design is appropriate, with methodology adequately described, and results are clearly presented.

Regarding to section of conclusions, authors describe and synthesize results, including a figure. It is important to remember that conclusions are propositions that express new knowledge, describing and explaining the behavior of the real problem assumed as the object of study. They are not propositions that comment, recommend, describe activities or synthesize results. For this reason, it is recommended to rewrite this section.

Finally, the manuscript has a high similarity index (>30%). Therefore, it is necessary to work on the wording to reduce these values.

Author Response

Comments and Suggestions for Authors

The study is interesting, reporting the metabomics differences between Dahongpao mother three and Cutting Dahongpao, in terms of odor and taste characteristics. The study contributes to the development of knowledge regarding this plant species that is so important in Chinese culture.

The research design is appropriate, with methodology adequately described, and results are clearly presented.

Regarding to section of conclusions, authors describe and synthesize results, including a figure. It is important to remember that conclusions are propositions that express new knowledge, describing and explaining the behavior of the real problem assumed as the object of study. They are not propositions that comment, recommend, describe activities or synthesize results. For this reason, it is recommended to rewrite this section.

 A: Thanks to the reviewers. The authors have revised the conclusion section appropriately. Hopefully, it meets the requirements. Thanks again to the reviewing experts.

Finally, the manuscript has a high similarity index (>30%). Therefore, it is necessary to work on the wording to reduce these values.

A: Thanks to the reviewers. The authors have carefully revised it against the test report.

Reviewer 2 Report

Comments and Suggestions for Authors

foods-3004089

Title: Mysterious "Dahongpao mother tree"(Camellia sinensis), its tea quality in the end how?

The article is fascinating and well-presented in the introduction, material and methods, results and discussion, and conclusion. There was one small item that needed to be rectified.

 The term "flavor" was used instead of "odor". This was due to odor, which is a bad fragrance in sensory analysis.

Please include the citation for the HPLC condition of non-volatile analysis.

All figures should include a caption, such as "Figure 1. Volatile metabolome analysis of tea tree leaves. (a) is……………. (b) is...”

Comments on the Quality of English Language

Please check for typographical errors.

Author Response

Comments and Suggestions for Authors

foods-3004089

Title: Mysterious "Dahongpao mother tree"(Camellia sinensis), its tea quality in the end how?

The article is fascinating and well-presented in the introduction, material and methods, results and discussion, and conclusion. There was one small item that needed to be rectified.

The term "flavor" was used instead of "odor". This was due to odor, which is a bad fragrance in sensory analysis.

A: Thanks to the reviewers. The authors have made appropriate changes in some parts of the manuscript. In addition, in the research literature on odor characteristics of tea, the OAV of volatile compounds is usually used to convert their contribution to a certain odor characteristic, and the full name of OAV is Odor Activity Value. Therefore, “odor” is a common professional term. Thanks again for the expert heads up.

Please include the citation for the HPLC condition of non-volatile analysis.

A: Thanks to the reviewers. The authors have made appropriate additions.

All figures should include a caption, such as "Figure 1. Volatile metabolome analysis of tea tree leaves. (a) is……………. (b) is...”

 A: Thanks to the reviewers. It is possible that the description of the figure was lost during the formatting conversion of the manuscript and has been added by the authors.

Comments on the Quality of English Language

Please check for typographical errors.

A: Thanks to the reviewers. The authors have carefully examined the full text.